# JAC4 Protects from X-ray Radiation-Induced Intestinal Injury by JWA-Mediated Anti-Oxidation/Inflammation Signaling

**DOI:** 10.3390/antiox11061067

**Published:** 2022-05-27

**Authors:** Yan Zhou, Jingwen Liu, Xiong Li, Luman Wang, Lirong Hu, Aiping Li, Jianwei Zhou

**Affiliations:** 1Department of Molecular Cell Biology & Toxicology, Center for Global Health, School of Public Health, Nanjing Medical University, 101 Longmian Avenue, Nanjing 211166, China; zhouyan_1223@163.com (Y.Z.); anglemuyan@foxmail.com (J.L.); hllixiong@163.com (X.L.); hdlxdlwlm@126.com (L.W.); liaiping@njmu.edu.cn (A.L.); 2The Key Laboratory of Modern Toxicology, Ministry of Education, School of Public Health, Nanjing Medical University, 101 Longmian Avenue, Nanjing 211166, China; 3State Key Laboratory of Translational Medicine and Innovative Drug Development, Jiangsu Simcere Pharmaceutical Co., Ltd., Nanjing 210042, China; hulirong@simcere.com

**Keywords:** JAC4, abdomen body irradiation, oxidative stress, mitochondrial apoptosis, DNA damage, small intestine

## Abstract

Radiation-induced intestinal injury is one of the major side effects in patients receiving radiation therapy. There is no specific treatment for radiation-induced enteritis in the clinic. We synthesized a compound, named JAC4, which is an agonist and can increase JWA protein expression. JWA has been shown to reduce oxidative stress, DNA damage, anti-apoptosis, and anti-inflammatory; in addition, the small intestine epithelium showed dysplasia in JWA knockout mice. We hypothesized that JAC4 might exert a protective effect against radiation-induced intestinal damage. Herein, X-ray radiation models were built both in mice and in intestinal crypt epithelial cells (IEC-6). C57BL/6J mice were treated with JAC4 by gavage before abdominal irradiation (ABI); the data showed that JAC4 significantly reduced radiation-induced intestinal mucosal damage and increased the survival rate. In addition, radiation-induced oxidative stress damage and systemic inflammatory response were also mitigated by JAC4 treatment. Moreover, JAC4 treatment alleviated DNA damage, decreased cell apoptosis, and maintained intestinal epithelial cell proliferation in mice. In vitro data showed that JAC4 treatment significantly inhibited ROS formation and cell apoptosis. Importantly, all the above protective effects of JAC4 on X-ray radiation-triggered intestinal injury were no longer determined in the intestinal epithelium of JWA knockout mice. Therefore, our results provide the first evidence that JAC4 protects the intestine from radiation-induced enteritis through JWA-mediated anti-oxidation/inflammation signaling.

## 1. Introduction

Currently, more than 50% of cancer patients worldwide receive radiation therapy, which triggers ionizing radiation to kill tumor cells [1]. However, ionizing radiation often causes damage to normal tissue, resulting in radiation toxicity [2]. The gastrointestinal (GI) mucosa shows an early response to radiation because of its rapid and continuous proliferation and regeneration [3]. Radiation-induced intestinal injury seriously affects the treatment of patients with abdominal or pelvic tumors and reduces life quality of patients [4]. However, there is no specific medicine for radiation-induced intestinal injury in the clinic. Therefore, the development of a highly effective and specific protective agent for radioactive intestinal injury is an urgent, unmet need worldwide.

Intestinal epithelial cells exposed to radiation will lead to cell death due to oxidative stress, DNA fragmentation, apoptosis, etc., resulting in a radiation damage subsequent reaction [5]. The damage of epithelial cells not only affects the villus–crypt structural integrity, but also compromises the barrier integrity and enhances intestinal permeability, allowing enteric bacteria to invade the damaged tissue and cause intestinal inflammation [6].

The JWA gene, also known as ADP ribosylation factor like GTPase 6 interacting protein 5 (ARL6IP5) (GenBank: AF070523, 1998), was originally identified and cloned from human bronchial epithelial (HBE) cells induced by all-trans retinoic acid [7]. Subsequent studies have shown that JWA is an active environmental response gene encoding a novel cytoskeletal binding protein that is involved in responding to oxidative stress, protecting normal cells from DNA damage through the base excision repair (BER) pathway [8], inhibiting inflammation [9] and anti-apoptosis [10]. Recently, we found that JWA-deficient mice exhibit intestinal epithelial dysplasia throughout life from the embryonic stage (which will be reported elsewhere). Therefore, we speculated that JWA may play an important role in the development and homeostasis of small intestinal epithelium in mice.

Small-molecule compounds have always been a hot spot in the research and development of new drugs [11]. JAC4 (JWA Agonist Compound 4) is a small-molecule compound screened by the high-throughput reporter gene assay in cooperation with the Chinese National Compound Library. JAC4 has been confirmed to transcriptionally activate JWA expression in several normal epithelial and cancer cells (which will be reported elsewhere). Although some studies have reported potential radiation protective agents, so far, no ideal radiation protective agent has been used in clinic [12]. In the present study, we report, for the first time, that JAC4 alleviated intestinal injury and reduced mortality in mice after X-ray exposure; JAC4 exerted both preventive and therapeutic roles on ABI-triggered intestine epithelial injury through elevating JWA expression. JAC4 will be an effective irradiation protecting agent against radiation-induced intestinal damage and inflammation.

## 2. Materials and Methods

### 2.1. JAC4 Synthesis and Preparation

JAC4 (C_16_H_13_N_3_O_3_S, MW: 327.4) was synthesized in our laboratory, and the purity was more than 98%. This compound was evaluated for acute toxicity. The median lethal dose (LD_50_) of JAC4 was more than 5000 mg/kg body weight in both male and female mice. Moreover, there were negative results in both the *Ames* test and mice erythrocyte micronucleus test for JAC4. In the present study, the solution was made by a prescription of polyethylene glycol (40%), ethanol (7.5%), and saline (52.5%) according to the volume, and the vehicle control was made by a compound-free solution.

### 2.2. Animals and Irradiation

Animals and X-ray radiation: male C57BL/6 mice (10 weeks old, 25 g) were provided by SLAC Laboratory Animal Center (Shanghai, China). The JWA conditional knockout (JWA^flox/flox^) mouse was contract constructed by the Model Animal Research Center of Nanjing University (Nanjing, China), for which the defined strain name of the mouse (B6.129 × 1-Arl6ip5tm1.2Jwzh, MGI ID_MGI:5437506) is now available on Mouse Genome Informatics (MGI). As described in our previous study [13], the exon2 of the JWA gene was flanked by two Loxp sites. The Villin-Cre mouse (B6. Cg-Tg (Vil1-cre)1000 Gum/J, RRID: IMSR_JAX:021504) was purchased from the Jackson Laboratory (Bar Harbor, ME, USA). The intestinal epithelial JWA deletion (Genotype: JWA^flox/flox^; Villin-cre, abbreviated as JWA^IEC-ko^) and the littermate wild-type (Genotype: JWA^flox/flox^, abbreviated as JWA^IEC-wt^) mice were obtained by mating JWA^flox/flox^ with Villin-Cre mice. After the Villin-Cre-mediated recombination, the exon2 of the JWA gene in the intestinal epithelial cells was deleted (Appendix A). The genotypes of the offspring were defined by PCR using the primers in Appendix A. The genotype of the JWA^IEC-ko^ mouse was identified with 437 bp (Appendix A, lanes 1 and 2) and 182 and 150 bp (Appendix A, lanes 2 and 3) fragments; the JWA^IEC-wt^ mice were identified with 437 bp (Appendix A, lanes 1 and 2) and 182 bp (Appendix A, lanes 1 and 4) bands. The use of the animals was approved by the Institutional Animal Care and Use Committee of Nanjing Medical University, Nanjing, China (IACUC-2004044). The mice were housed in a specific pathogen-free (SPF) environment with controlled humidity and temperature and a 12/12 light/dark cycle with free access to drinking water and standard chow. For the survival experiment model, the mice (n = 10 per group) were exposed to ABI at a single lethal dose of 12 Gy and then monitored for 30 days. JAC4 (100 mg/kg body weight/day) or an equal volume vehicle was orally administered 7 days before and 3 days after ABI.

The dose–response mice model: with regard to the radiation challenge to the small intestine, mice (n = 6 per group) were exposed to ABI at a dose of 3/6/12 Gy, respectively. The small intestine was harvested for the histopathology evaluation after 6 h, 4 d, or 8 d. JAC4 (100 mg/kg body weight/day) or an equal volume vehicle was orally administered 7 days before and 4 days after ABI.

The irradiation model of the intestinal epithelial JWA knockout mice (Off-target model): JWA^IEC-ko^ mice and littermate JWA^IEC-wt^ mice (8 weeks old) were used in the Off-target model. The irradiation-exposed mice were divided into four groups: (1) JWA^IEC-wt^ mice + vehicle group (n = 5); (2) JWA^IEC-wt^ mice + JAC4 group (n = 5); (3) JWA^IEC-ko^ mice + vehicle group (n = 6); (4) JWA^IEC-ko^ mice + JAC4 group (n = 5). All the mice were exposed to ABI (10 Gy). Then, 10 mg/kg JAC4 or an equal volume vehicle intragastric administration was performed 7 days before irradiation and continued until the end of the model. Mice in the blank control group (wild-type, n = 6) were not irradiated and the model was terminated on the fourth day after irradiation.

All the X-ray radiation exposure was performed at the Animal Core Facility of Nanjing Medical University, with an Rs-2000 Pro X-ray irradiator (Rad Source Technologies, Buford, GA, USA) at a dose rate of 1.25 Gy/min. The irradiation range was 3 cm above the iliac joint.

### 2.3. Cell Culture and Irradiation

The rat small intestinal epithelial line cell IEC-6 (CRL-1592) was purchased from ATCC (MD, Manassas, VA, USA) and cultured in Dulbecco’s modified eagle medium (DMEM) supplemented with 100 μg/mL streptomycin, 100 U/mL penicillin, and 10% fetal bovine serum in an incubator with 5% CO_2_ at 37 °C. The IEC-6 cells were plated and, after adhesion, were pretreated with JAC4 (1 or 10 µM) for 24 h and then exposed to the X-ray irradiation (12 Gy) at a dose rate of 1.25 Gy/min, followed by relevant cell biology experiments 24 h later.

### 2.4. Histopathological Score of the Small Intestine

All the intestinal segments of the mice were removed immediately after euthanasia, and the small intestine was collected and rinsed with cold saline. The fragments were placed in 4% paraformaldehyde solution, fixed for 24 h, and treated into bundles or “Swiss roll” form. The processed small intestine tissue was embedded in paraffin and cut into 5 μm sections. Hematoxylin and eosin (H&E) staining was conducted using an H&E staining kit according to the manufacturer’s protocol (Servicebio, Wuhan, China).

The degree of radiation damage in mice was evaluated, as previously described [14,15]. Histological abnormalities were graded as follows: Grade 0: normal; Grade 1: inflammatory cell infiltration or mucosal bleeding; Grade 2: vacuolization of villi, abnormal crypt direction, or mucosal hypertrophy; Grade 3: submucosal cysts or irregular crypt regeneration of atypical epithelial cells; Grade 4: ulcerative mucosal or transmural necrosis. Grade 3 or 4 slides were considered to have severe histological damage.

A blind and independent objective analysis and quantification of crypt depth on H&E-stained slides were performed using Image J software. Measuring the height from the base of the crypt to the junction of the crypt villi, mucosal villus height was measured from the tip to the base of each villus, and crypt depth was measured from the crypt base to the top opening. The average villi height was measured from different locations in the small intestine of at least three different animals in each group. Data were reported as mean ± standard deviation.

### 2.5. Intestinal Permeability Assay

Mice were fasted for 12 h with ad libitum drinking. On the fourth day after radiation, animals were given a FITC-Dextran 4 kDa intragastric administration at the dose of 0.6 mg/g body weight. Peripheral blood plasma was collected 3 h later and separated in an anticoagulant tube, followed by 8000 RCF centrifugation for 10 min. Then, the sample was mixed with PBS in a 1:1 volume ratio, and a 100 μL sample was added to each well. The fluorescence intensity was measured with an enzyme standard instrument (TECAN, Hombrechtikon, Männedorf, Swiss) at an excitation wavelength of 485 nm and an emission wavelength of 535 nm in a 96-well plate. Standard curves were prepared at concentrations of 0, 12.5, 25, 37.5, 50, 75, 100, 200, 400, 600, and 800 μg/mL. The permeability of each sample was calculated according to the standard curve.

### 2.6. Western Blotting Analysis

The IEC-6 cells or intestinal tissue samples were prepared with RIPA lysis buffer containing a protease inhibitor cocktail (Millipore, Danvers, MA, USA). The protein concentration in the lysate was determined with a bicinchoninic acid assay (BCA) kit (Beyotime, Shanghai, China). Equal amounts of protein were separated by SDS-PAGE and electroblotted on PVDF membrane. After blocking with 5% non-fat dry milk for 1 h, the membranes were incubated with the specific primary antibodies overnight at 4 °C. After washing with TBST (6 min × 4), the membranes were incubated with a horseradish peroxidase (HRP)-linked second antibody for 1 h at room temperature. After washing (8 min × 4), the blots were detected by an enhanced chemiluminescence (ECL) Western blotting detection kit (Beyotime). Densitometry analyses were performed with ImageJ. β-actin was used as the loading control marker. The following antibodies were used in the present study: anti-JWA (1:100, AbMax, Beijing, China), anti-XRCCI (1:1000, Cell Signaling Technology, Danvers, MA, USA), anti-caspase-3 (1:1000, Cell Signaling Technology, Danvers, MA, USA), anti-caspase-9 (1:1000, Cell Signaling Technology), anti-Bax (1:1000, Cell Signaling Technology), anti-Bcl2 (1:1000, Cell Signaling Technology), anti-ZO-1 (1:1000, Proteintech, Wuhan, China), anti-Claudin-3 (1:1000, Proteintech), anti-cleaved-PARP1 (1:1000, Cell Signaling Technology), and anti-β-actin (1:1000, Beyotime).

### 2.7. Intracellular ROS Assay

The content of intracellular ROS was measured using the ROS detection kit (Beyotime). Briefly, after a specific experimental treatment, the culture medium was discarded and then incubated with 10 μM of DCFH-DA. After an incubation of 30 min at 37 °C, cell fluorescence intensity was observed under a fluorescence inverted microscope (Nikon, Tokyo, Japan).

### 2.8. Flow Cytometric Assay

A total of 2 × 10^5^ IEC-6 cells were plated in six-well plates and treated with JAC4 (1 μM) or DMSO for 24 h prior to X-ray irradiation exposure at 12 Gy. Twenty-four hours after the treatment, cells were harvested in PBS and analyzed with an Annexin V Apoptosis Detection kit (Beyotime) with a flow cytometer (BD, Franklin Lakes, NJ, USA).

### 2.9. Biochemical Analysis

The mice plasma was detected in strict accordance with the instructions of malondialdehyde (MDA), glutathione peroxidase (GSH-PX), superoxide dismutase (SOD), catalase (CAT), and DAO kits (Jiancheng Bioengineering Institute, Nanjing, China). The BCA method was used to measure the intracellular protein concentration, and the results were obtained after standardization. At least three biological replications were conducted for all experiments.

### 2.10. Hoechst Staining

Apoptosis was determined by Hoechst 33342 staining (Beyotime). A total of 2 × 10^5^ IEC-6 cells were plated in six-well plates. After treatment with JAC4 (1 μM) for 24 h, the cells were fixed with 4% paraformaldehyde and then stained with the Hoechst 33342 solutions for 20 min at 37 °C. Images were acquired under a laser scanning confocal microscope (Carl Zeiss, Jena, Germany).

### 2.11. Measurement of Mitochondrial Membrane Potential

Mitochondrial membrane potential was measured by the cationic dye JC-1 (Beyotime). Briefly, after treatment and washing three times with PBS, culture medium (1 mL) and JC-1 staining solution (1 mL) were added. After being fully mixed, the cells were incubated at 37 °C for 20 min in the incubator. During incubation, an appropriate amount of JC-1 staining buffer (1×) was prepared at the ratio of 4 mL distilled water for every 1 mL JC-1 staining buffer (5×) and placed in an ice bath. After incubation at 37 °C, the supernatant was removed and washed twice with JC-1 dyeing buffer (1×). For measurement, a fluorescence microscope (Carl Zeiss) was used for viewing the fluorescence intensity, and flow cytometer (BD) was used for quantifying the ratios of red/green fluorescent intensities.

### 2.12. TNF-α, IL-1β, and IL-10 Assay

Mice plasma was collected for the detection of the levels of TNF-α, IL-1β, and IL-10. These cytokines were measured by using a commercial ELISA kit (Proteintech) according to the manufacturer’s instructions. The levels of TNF-α and IL-1β in the plasma was expressed as nanograms per liter and IL-10 levels was expressed as picograms per liter.

### 2.13. Immunofluorescence Assay

For immunofluorescence detection, sections and cells were incubated with primary antibody and then labeled with an anti-rabbit secondary antibody conjugated with Cy3 or FITC (1:1000) in the dark for 1 h at room temperature. Cell nuclei were counterstained with 40, 60-diamidino-2-phenylindole (DAPI). Images were acquired under a laser scanning confocal microscope (Carl Zeiss Jena) or fluorescence inverted microscope (Nikon). The following antibodies were used: anti-caspase-3 (1:2000, Cell Signaling Technology), anti-ZO-1 (1:1000, Proteintech), anti-Oculdin-3 (1:2000, Proteintech), anti-γ-H2AX (1:1000, Cell Signaling Technology), and anti-Ki-67 (1:1000, Cell Signaling Technology).

### 2.14. The Cytochrome C Release Assay

The IEC-6 cells were seeded at a density of 5 × 10^4^ per well in 24-well culture plates with a cell lamella. After attachment, the cells were cultured with JAC4 for another 24 h. After incubation, the cells were irradiated using the X-ray (12 Gy). After a 24 h incubation, Mito-tracker (Beyotime) was employed to stain the mitochondria for 30 min, and then the IEC-6 cells were fixed and punched. Next, the cells were specifically blocked for 1 h and then treated with cytochrome c mouse antibody (Beyotime) at room temperature for 3 h. Then, the cells were cultured with a fluorescent secondary antibody for another 1 h. Finally, the cells were imaged by a laser scanning confocal microscope (Carl Zeiss).

### 2.15. Statistics Analysis

All statistical analyses were performed with GraphPad Prism 8 software and/or SPSS 25.0. The data were expressed as the means ± S.D. Multiple means were compared by ANOVA and pairwise comparisons were analyzed with the Student’s *t*-test. The Chi-square test was used to compare counting data. Survival data were plotted using the Kaplan–Meier method, and the differences between groups were analyzed using the log-rank test; *p* < 0.05 was considered statistically significant.

## 3. Results

### 3.1. JAC4 Improves the Survival Rate and Reduces the Damages of Intestinal Epithelium after ABI in Mice

To determine the potential protective effects of JAC4 on irradiation in mice, we firstly observed the survival rates of mice after 12 Gy ABI. The C57BL/6J mice (10 weeks old, 25–30 g) were treated with 100 mg/kg JAC4 or vehicle for 7 days before exposure and 3 days post-exposure to 12 Gy ABI, which was followed by monitoring survival for 30 days (Figure 1A). The results revealed that all the mice (10/10) in the vehicle-treated group died within 15 days post ABI, while 6/10 mice in the JAC4-treated group were alive on the 15th day post ABI and 2/10 of mice were still alive on the 30th day, the end point of the experiment (Figure 1B). After ABI, the average survival was 8.3 days in the vehicle group, while the JAC4 (100 mg/kg/day)-treated group had an average survival of 16.8 days (Figure 1C, *p* < 0.05). The body weight loss was also less in the JAC4-treated mice than that of the vehicle group after ABI (Appendix A). JAC4 also increased the survival of mice after 6.5 Gy total body irradiation (TBI) (Appendix A); at the same time, the blood biochemical results showed that the JAC4 treatment reduced the increase in ALT, AST, LDH, and CK-MB after TBI (Appendix A), suggesting that JAC4 reduced the organ toxicity of the heart and liver by radiation.

To elucidate if the radioprotective role of JAC4 in mice was by reducing intestinal epithelial damage, the mice were irradiated at 3, 6, or 12 Gys and evaluated for morphological changes of the small intestine on the fourth day after ABI. Data showed 3, 6, or 12 Gy irradiation resulted in 31.5% (*p* < 0.001), 40.7% (*p* < 0.001), and 50.2% (*p* < 0.001) villi shortening in the mice, respectively (3 Gy vs. 6 Gy *p* < 0.05, 6 Gy vs. 12 Gy *p* < 0.05); furthermore, the crypt height was also decreased by 77.7% (*p* < 0.001), 73.2% (*p* < 0.001), and 68.9% (*p* < 0.001) in the mice, respectively (Figure 1D–F, 3 Gy vs. 12 Gy *p* < 0.01).

To determine whether JAC4 plays a dual role in alleviating ionizing radiation-induced intestinal epithelial injury and simultaneously promoting repairing or regeneration, we treated C57BL/6 mice with either vehicle or JAC4 (100 mg/kg/day) 7 days before-exposure and 4 days post-exposure of 12 Gy ABI and analyzed the morphological changes of the small intestine at 6 h or 8 days after ABI (Figure 1G). The pathological results showed severe small intestine epithelium damage in the vehicle-treated mice at 6 h after ABI; however, the mice pretreatment of JAC4 indicated mild impairment of the small intestine epithelium (Figure 1H). Similarly, on the eighth day after ABI, compared to the severely damaged small epithelium in the control group mice, the JAC4-treated mice showed almost completely repaired small intestine epithelium (Figure 1H). The relative quantitative data indicated that at both 6 h and 8 days after ABI, the villi height was lower in the vehicle group than in the JAC4-treated mice (Figure 1I, *p* < 0.001). The crypt height was lower in the vehicle group at 6 h after ABI than in the JAC4-treated mice; however, there was no significant difference between the two mice groups at 8 days after ABI (Figure 1J, *p* < 0.05). The histopathological scores were obviously lower in the JAC4-treated mice than in the vehicle group at both 6 h and 8 days after ABI, respectively (Figure 1K, *p* < 0.001). In addition, the histopathological scores in the JAC4-treated mice were obviously higher than that of the non-ABI mice (Figure 1K, *p* < 0.001). In order to clarify the protective effect of JAC4 on radiation-induced intestinal epithelial damage, we carried out preventive intervention of JAC4 7 days before TBI and therapeutic intervention of JAC4 7 days after TBI, respectively. The analysis of the histopathological results of the small intestine showed that JAC4 improved the rupture and length shortening of the small intestinal villi caused by irradiation, whether preventive administration or therapeutic administration (Appendix A).

### 3.2. JAC4 Improves the Intestinal Mucosal Barrier from Radiation Damage in Mice

The intestinal mucosal barrier is one of the important immune barriers of the body. To determine if JAC4 improves the intestinal epithelial barrier in mice after ABI, we detected the expressions of the tight junction protein ZO-1 and Claudin-3 in the ABI model mice. As shown in Figure 2A,B, X-ray irradiation (12 Gy, 4 days post-ABI) obviously reduced the fluorescence intensities of both ZO-1 and Claudin-3; however, JAC4 treatment prevented the reduction in both ZO-1 and Claudin-3. Similarly, the data of Western blotting showed that the expressions of both ZO-1 and Claudin-3 were downregulated slightly, and JAC4 reversed the expressions of ZO-1 and Claudin-3 (Figure 2C–E). Next, we determined the FITC-dextran and DAO contents in the plasma of mice 4 days after X-ray ABI treatment for the evaluation of the integrity of the small intestine barrier. As expected, compared to the non-ABI mice, plasma FD4 and DAO levels were increased after ABI in mice (*p* < 0.01, *p* < 0.001, respectively); however, compared to the vehicle-treated group, JAC4 treatment reduced both the levels of FITC-dextran (*p* < 0.05) and DAO (*p* < 0.01) in the plasma (Figure 2F,G). These results indicated that JAC4 effectively ameliorated the irradiation-destructed intestinal mucosal barrier in mice.

### 3.3. JAC4 Inhibits X-ray-Triggered Inflammation and Oxidative Stress

Radiation is a multifactorial pathological lesion that causes inflammation of the small intestine, manifested as a shortening of length due to edema of the small intestine. To investigate whether the protective roles of JAC4 on ABI-induced intestinal injury were through reduction in intestinal inflammation, we completed the animal model shown in Figure 3A. The results indicated that, compared to the non-irradiation-treated mice, X-ray irradiation (12 Gy) obviously shortened the average length of the small intestine in mice (36.4 ± 2.2 vs. 31.4 ± 3.0 cm, *p* < 0.01); however, JAC4 treatment improved the length of the small intestine of ABI mice (31.4 ± 3.0 vs. 34.8 ± 1.4 cm, *p* < 0.05) (Figure 3B,C) while there were no significant changes in colon length between vehicle- and JAC4-treated ABI mice (Figure 3D). We further detected serum levels of pro-inflammatory factors including IL-1β, TNF-α, and the anti-inflammatory factor IL-10. The results showed that serum IL-1β, TNF-α, and IL-10 were increased after ABI in mice. However, JAC4 treatment obviously prevented the increase in both IL-1β and TNF-α, and further enhanced the serum IL-10 level in mice (Figure 3E–G), suggesting that JAC4 alleviates intestinal inflammation caused by X-ray radiation.

Given that the irradiation-triggered small intestine inflammation was due to oxidative stress events, we detected the anti-oxidant status of the intestine by the contents of the end product of lipid peroxidation malondialdehyde (MDA) and the activities of glutathione peroxidase (GSH-Px), superoxide dismutase (SOD), and catalase (CAT) in the serum of mice. As shown in Figure 3H, MDA contents were increased by ABI while obviously being reduced by JAC4 in the mice. As a response to oxidative stress, ABI increased the activities of GSH-Px, CAT, and SOD; these anti-oxidative enzymes were further enhanced by JAC4 treatment (Figure 3I–K). These data support that JAC4 inhibited small intestine inflammation and was associated with its anti-oxidant function.

### 3.4. JAC4 Attenuates X-ray-Triggered DNA Damage and Apoptosis of the Small Intestinal Epithelial Cells in Mice

Both ionizing radiation and oxidative stress can cause DNA damage and apoptosis of target organ cells. We detected the biomarkers of both DNA damage and apoptosis in the small intestine of the ABI mice. The immunofluorescence staining data demonstrated that, compared to the non-irradiation group, X-ray exposure significantly increased the γH2AX levels on the fourth day post-irradiation in mice; however, JAC4 treatment obviously reduced the γH2AX levels of the mice (Figure 4A). To understand the effect of JAC4 on intestinal epithelial cell proliferation, we performed Ki67 staining in the small intestine tissue from ABI mice. Data showed that ABI significantly reduced the number of Ki67 positive intestinal epithelial cells on the fourth day, while JAC4 treatment prevented the effects of ABI and indicated almost normal cell proliferation in intestine epithelium (Figure 4B). Moreover, caspase-3 staining demonstrated that ABI-increased apoptosis in the small intestine epithelium was obviously reduced by JAC4 treatment (Figure 4C). To confirm the mechanisms by which JAC4 protects against a radiation-induced imbalance between cell proliferation and apoptosis, we determined the expressions of related protein biomarkers by Western blot. As shown in Figure 4D, X-ray exposure increased the expressions of Bax, cleaved caspase-3, and cleaved PARP1 in the small intestine epithelium, which were reversed by JAC4; more importantly, JAC4 treatment also increased Bcl2 expressions except for JWA (Figure 4D–I). Taken together, these findings suggested that JAC4 protected the small intestine from X-ray-induced DNA damage and maintained the balance between cell apoptosis and anti-apoptosis signaling after ABI.

### 3.5. JAC4 Protects ICE-6 Cells from X-ray-Triggered DNA Damage/Apoptosis through Mitochondria Signaling

We next investigated the protective potential of JAC4 against X-ray-induced damage in IEC-6 cells to verify its mechanisms. The IEC-6 cells were pretreated with JAC4 (1 μM) for 24 h and then exposed to X-ray (12 Gy); the relevant experiments were performed 24 h later. Results showed that X-ray exposure significantly increased intracellular levels of ROS and nuclear γ-H2AX; however, JAC4 pretreatment obviously reduced the levels of both ROS and γ-H2AX (Figure 5A,B). Moreover, X-ray exposure elevated the expressions of JC-1 on the mitochondria membrane and apoptosis rate in the IEC-6 cells (*p* < 0.001). The pretreatment of JAC4 protected ICE-6 cells from X-ray-triggered mitochondria damage and apoptosis (*p* < 0.01, Figure 5A,C,D). In addition, we evaluated the cell apoptosis (Annexin V staining) by flow cytometry. As shown in Figure 5E,F, compared to the control cells, X-ray exposure increased early apoptotic cells (4.49% vs. 10.3%, *p* < 0.001); however, JAC4 pretreatment protected cells from X-ray-triggered early apoptosis (10.3% vs. 5.57%, *p* < 0.01). The immunofluorescence staining also showed X-ray exposure increased cytoplasmic cytochrome c and it was translocated into cell nuclei upon induction of DNA damage [16]; JAC4 protected mitochondria and reduced cytochrome c releasing (Figure 5H). The Pearson’s correlation assay showed X-ray exposure decreased the colocalizations between mito-tracker and cytochrome c in ICE-6 cells; however, JAC4 pretreatment reversed the colocalizations of both biomarkers (Figure 5G).

To further confirm the mechanisms by which JAC4 protects cells from X-ray exposure-triggered mitochondria damage, we detected related biomarkers by Western blot assay. As shown in Figure 5I, JAC4 treatment increased JWA expression in ICE-6 cells, and 1 μM JAC4 showed better effect than 10 μM. X-ray exposure led to the reduction in XRCC1, although increasing the levels of both cleaved caspase-3 and cleaved PARP1. In line with the above data, JAC4 treatment increased expressions of XRCC1 and Bcl-2, although decreasing the expressions of cleaved caspase-9/-3/PARP1 and Bax. The protecting effects of JAC4 were indicated with dose-dependent manners. These results suggested that JAC4 protected cells from X-ray-triggered DNA damage/apoptosis through mitochondria-related signaling.

### 3.6. JAC4 Protects Mice from X-ray-Induced Intestinal Barrier Damage Dependent upon Intestinal Epithelium JWA Expression

To verify if JAC4 protects mice from X-ray irradiation-induced intestinal injury depend upon JWA expression. The JWA^IEC-ko^ mice were used to verify the targeted effects of JAC4. In the irradiated intestinal injury model, we further studied the dose–response relationship of JAC4. C57BL/6 mice were treated with either a vehicle or JAC4 (10/30/100 mg/kg/day) 7 days before exposure and 4 days post-exposure to 12 Gy ABI. We ended the model on the fourth day after ABI. Results showed that 10 mg/kg JAC4 improved the small intestine length shortening and reduced the serum FD4 concentration after ABI and showed better intestinal protection after irradiation than 100 mg/kg (Appendix A). Therefore, we used a dose of 10 mg/kg JAC4 in the Off-target mouse model. We also conducted a wild-type mouse irradiation model for comparison, with 10 mg/kg of JAC4. Both the JWA^IEC-ko^ and JWA^IEC-wt^ mice were treated with either a vehicle or JAC4 (10 mg/kg/day) 7 days before and 4 days post-exposure to ABI (10 Gy) (Figure 6A). The plasma biomarkers including intestinal barrier (FD4), inflammation (TNF-α, IL-1β), and oxidative stress (CAT, GSH-px) were determined. We found that, compared with the blank control group (4758.4 ± 1066.0 ng/mL), the FD4 content in the vehicle group (12,865.5 ± 5539.5 ng/mL) was significantly increased after irradiation; however, FD4 content was significantly decreased (6719.9 ± 1823.5 ng/mL) after JAC4 treatment, suggesting that 10 mg/kg JAC4 can reduce irradiation-induced intestinal damage in JWA^IEC-wt^ mice (*p* < 0.05). Unfortunately, in the JWA^IEC-ko^ mice, there was no significant difference in the FD4 content after JAC4 treatment (9192.8 ± 5573.2 vs. 7673.7 ± 1474.2 ng/mL), suggesting that JAC4 did not protect from irradiation-induced intestinal damage in the JWA^IEC-ko^ mice (Figure 6B). Meanwhile, the contents of TNF-α and IL-1β were shown to be consistent with the FD4 results. Suggest 10 mg/kg JAC4 could reduce the inflammatory response after irradiation in the JWA^IEC-wt^ mice (TNF-α: 583.2 ± 78.2 vs. 783.5 ± 192.6 ng/mL; IL-1β: 57.76 ± 5.9 vs. 66.7 ± 7.5 ng/mL) (*p* < 0.05) but did not show similar effects in the JWA^IEC-ko^ mice (TNF-α: 769.8 ± 112.3 vs. 668.8 ± 112.7 ng/mL; IL-1β: 65.6 ± 3.3 vs. 66.7 ± 7.5 ng/mL) (Figure 6C,D). As shown in Figure 6E,F, ABI increased the activity of GSH-Px (530.8 ± 207.8 vs. 406.2 ± 197.9 U/mL) and decreased the activity of CAT (89.4 ± 0.4 vs. 68.7 ± 14.2 U/mL) (*p* < 0.01). These anti-oxidative enzymes were further enhanced by JAC4 treatment in the JWA^IEC-wt^ mice (815.1 ± 203.9 vs. 530.8 ± 207.8 U/mL and 85.9 ± 8.2 vs. 68.7 ± 14.2 U/mL) (*p* < 0.05, *p* < 0.05, respectively); however, these results were not confirmed in the JWA^IEC-ko^ mice. Histological analysis confirmed intestine injury in the normal control mice on day 4 after irradiation and showed severe shortening of the villi length (166.3 ± 14.4 vs. 240.7 ± 17.4 μm) and crypts (55.3 ± 3.7 vs. 76.7 ± 10.0 μm), which was improved in the JAC4-treated mice (villus: 180.6 ± 9.4 vs. 166.3 ± 14.4 μm; crypt: 68.6 ± 6.6 vs. 55.3 ± 3.7 μm). However, in the JWA^IEC-ko^ mice, JAC4 did not improve intestinal tissue injury (villus: 150.6 ± 8.4 vs. 150.0 ± 11.8 μm; crypt: 53.0 ± 3.0 vs. 51.0 ± 3.8 μm) (Figure 6G–I). We isolated the intestine tissue and evaluated the expression of apoptotic molecules by Western blot assay (Figure 6J). In wild-type mice, X-ray increased the expression of Bax and cleaved PARP1 in the small intestine compared with the control group. In contrast, mice treated with JAC4 downregulated the expression of Bax and cleaved PARP1 but upregulated that of the anti-apoptotic molecule Bcl2, which was not observed in the JWA^IEC-ko^ mice after JAC4 treatment. The results suggest that JAC4, through the activation of JWA, protected the intestinal epithelium from injury by X-ray irradiation exposure in mice. In the JWA^IEC-ko^ mice, JAC4 could not alleviate the injury of the intestinal epithelium after irradiation by targeting JWA activation.

## 4. Discussion

Gastrointestinal (GI) complications are a major, dose-limiting factor in patients receiving abdominal radiation [17]. There is no radiation protective drug available for the treatment of acute gastrointestinal radiation syndrome [18]. In the present study, we investigated the protective effects of the JWA small-molecule agonist JAC4 on radiation-induced intestinal injury. Varying degrees of villi shortening or loss and crypt atrophy may occur after radiation exposure, leading to the disruption of epithelial homeostasis and epithelial integrity [19]. Intestinal mucosal injury can lead to nutrient deficiencies, weight loss, and even death [20]. Our results showed that JAC4 treatment slowed weight loss and reduced death in mice, and the intestinal crypt villi structure of mice was preserved after 12 Gy ABI. JAC4 also improved the survival of mice and protected against cardiohepatic toxicity in mice after total body irradiation (TBI). Furthermore, both prophylactic and therapeutic administration of JAC4 improved the rupture and shortening of the small intestinal villi induced by TBI.

Radiation-induced changes in tight junction-related protein levels play a key role in enhancing intestinal permeability. Claudin-3 and ZO-1 are major components of tight junction-related proteins, which are essential for maintaining the integrity of the intestinal epithelial barrier [21]. Notably, compared with vehicle treatment, JAC4-treated mice significantly increased the expression of Claudin-3 and ZO-1 and reduced the levels of serum DAO and FD4 after ABI, suggesting that JAC4 may help restore barrier integrity by Occludin-3 and ZO-1. All these data clearly indicate that JAC4 has a protective effect on radiation-induced intestinal epithelial injury.

Radiation exposure generates excessive ROS-caused oxidative stress damage [22]. JWA reduces oxidative stress in normal cells induced by hydrogen peroxide through the activation of anti-oxidant enzymes and the base excision repair (BER) pathway [8] and the inhibition of inflammation [9]. The use of JAC4 obviously enhanced the anti-oxidant capacity in X-ray-exposed mice and cells and, therefore, maintained the oxidation/reduction balance. As a consequence, JAC4 treatment inhibited the pro-inflammatory factor IL-1 β and increased the level of anti-inflammatory factor IL-10.

Studies have shown that radiation-induced cell apoptosis increases tissue damage [23]. Caspase-3 is classified as executioner caspases by the mechanism of action [24]. In the present study, JAC4 reduced the number of apoptotic cells in the small intestine epithelium after ABI by inhibiting caspase-3 expression; at the same time, JAC4 increased the expression of Ki67, a proliferative marker in the small intestine. The phosphorylated H2AX is a variant form of histone H2A, which has been widely used as a marker for DNA double-strand breaks [25]. In this study, we observed that the expression of γH2AX of small intestine epithelium was decreased in JAC4-treated mice after ABI. Proteins of the Bcl-2 family are either anti-apoptotic (Bcl-2-like proteins, such as Bcl-2 and Bcl-x) or pro-apoptotic (for example, Bax and Bak) [26]. Bcl-2 inhibits apoptosis by stabilizing cell membrane permeability and maintaining mitochondrial integrity [27], while Bax increases membrane permeability and results in the release of Cyto-C from mitochondria into the cytoplasm, ultimately leading to cell apoptosis [28]. Thus, the homeostasis and balance between anti-apoptotic Bcl-2 and pro-apoptotic Bax (Bcl-2/Bax) is a decisive factor in determining the fate of cell apoptosis. In this study, both in vitro and in vivo evidence showed that JAC4 treatment significantly reversed the radiation-induced imbalance of Bcl-2/Bax molecules and prevented the mitochondrial transmembrane potential loss, which greatly reduced the release of Cyto-C from mitochondria. As a result, JAC4 effectively inhibited the activation and shearing of caspase-3 and caspase-9 and ultimately reduced cell apoptosis induced by radiation.

JAC4 is a small-molecule compound that transcriptionally activates JWA gene expression and is screened by the high-throughput reporter gene assay from the National Compound Library. To verify that JAC4 protection from intestinal injury after irradiation is by targeting JWA, we constructed an intestinal epithelium JWA knockout mice ABI model. The results showed that JAC4 failed to improve the intestinal barrier injury, oxidative stress, and inflammation in intestinal epithelial JWA knockout mice, suggesting that the JWA gene is necessary for JAC4 against intestinal radiation injury.

Rapid renewal predisposes the GI epithelium to common side effects seen in cancer patients receiving radiotherapy [29]. It was reported that 80% of patients receiving abdominal radiotherapy develop intestinal mucositis caused by injuries to normal intestinal epithelial cells within a few weeks [30]. Patients with acute radiotherapy enteritis manifest symptoms, such as diarrhea, abdominal pain, and loss of weight [31]. Fortunately, here, we demonstrated that JAC4 is effective by virtue of increased survival, reduced injury, and enhanced recovery of intestinal epithelium in acutely irradiated animals. JAC4 is a safe small-molecule compound that can be administered orally in tablet and powder forms, which will improve patients’ compliance with JAC4. We think that JAC4 has potential clinical translational application value as a promising radioprotector, and we are also working with a pharmaceutical company to conduct preclinical studies. JAC4 is also indicated to improve mitochondrial energy metabolism reprogramming in pancreatic cancer cells by promoting oxidative phosphorylation and inhibiting glycolysis [32]. As the target gene of JAC4, JWA is reported as a tumor suppressor in several cancers such as gastric cancer [33,34], non-small cell lung cancer [35], breast cancer [36,37], melanoma [38,39], etc.; JAC4 combined with radiation therapy for cancer may play a potential role of killing two birds with one stone and indicating the potential translational significance.

Of course, the efficacy of JAC4 combined with X-ray radiation in cancer treatment as well as the types of cancers that can be treated and the mechanisms behind the action need to be further verified. In addition, both the animal and cell models designed in this study were acute injuries caused by high-dose X-ray; it remains to be studied whether JAC4 also has a protective effect on long-term, low-dose X-ray exposure. The pathophysiology of radiation-induced intestinal injury is related to the gut micro-flora dysregulation. Studies have proven that ABI disrupted the balance of intestinal microbiota and significantly reduced the diversity of intestinal microbiota in mice [40], reducing the survival of radioactive mice [41]; whether JAC4 has a regulatory effect on the intestinal flora balance needs further study.

## 5. Conclusions

This study demonstrated the anti-oxidation/inflammation potential of JAC4, a JWA agonist, in intestinal injury model mice induced by X-ray. The results showed that JAC4 reduced the serum level of pro-inflammatory cytokines (TNF-α and IL-1β) and oxidative stress (MDA) and enhanced the anti-inflammatory factor IL-10 and anti-oxidative enzymes (GSH-Px and CAT levels). JAC4 also contributed to the restoration of the tissue architecture of villi and crypts in the small intestine, improved the intestinal mucosal barrier, and improved clinical parameters such as survival, body mass variation, and blood biochemical index. Thus, the results of this study suggest that JWA is a potential target for future prevention and therapy of radiation-induced intestinal injury; the JWA gene agonist compound JAC4 will be one of the candidate radioprotection agents.

## Figures and Tables

**Figure 1 antioxidants-11-01067-f001:**
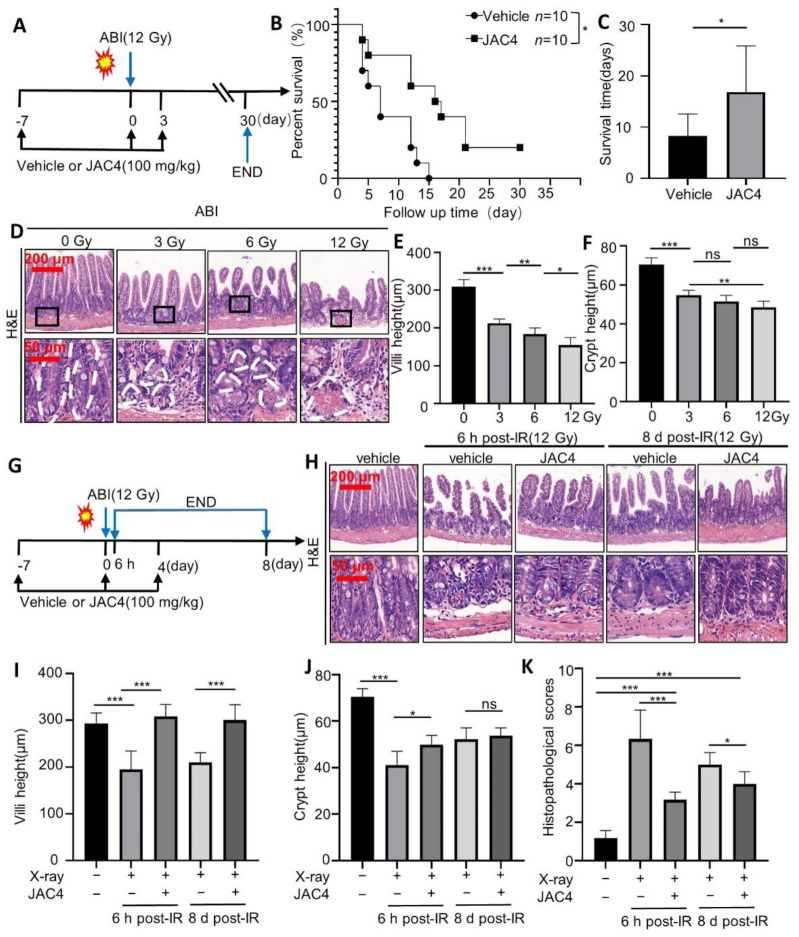
JAC4 improves the survival rate of mice and reduces the damages of intestinal epithelium after ABI. (**A**) C57BL/6 mice (n = 10) were administered JAC4 (100 mg/kg) or vehicle by gavage for 7 days. On the seventh day, mice were exposed to radiation (12 Gy). JAC4 treatment was continued for an additional 3 days and then monitored for 30 days. (**B**) Kaplan–Meier survival analysis of mice exposed to 12 Gy ABI (*p* < 0.05, n = 10 per group). (**C**) The average survival time of the two groups of mice (*p* < 0.05). (**D**–**F**) Hematoxylin and eosin (H&E) staining of jejunum and quantification of the villus height and crypt height. Mice (n = 6) were administered JAC4 (100 mg/kg) or vehicle by gavage daily for 7 days prior to 3/6/12 Gy ABI; JAC4 treatment was continued for an additional 4 days. (**G**) Mice (n = 6) were administered JAC4 (100 mg/kg) or vehicle by gavage for 7 days. On the seventh day, mice were exposed to radiation (12 Gy). JAC4 treatment was continued for four days. Then, serum and small intestine tissues were collected at 6 h and 8 d, respectively. (**H**) H&E staining in the jejunum. (**I**) Villus height, (**J**) crypt height, and (**K**) histopathological scores. The results are presented as mean ± SD (n = 6) (* *p* < 0.05; ** *p* < 0.01; *** *p* < 0.001; ns: *p* > 0.05).

**Figure 2 antioxidants-11-01067-f002:**
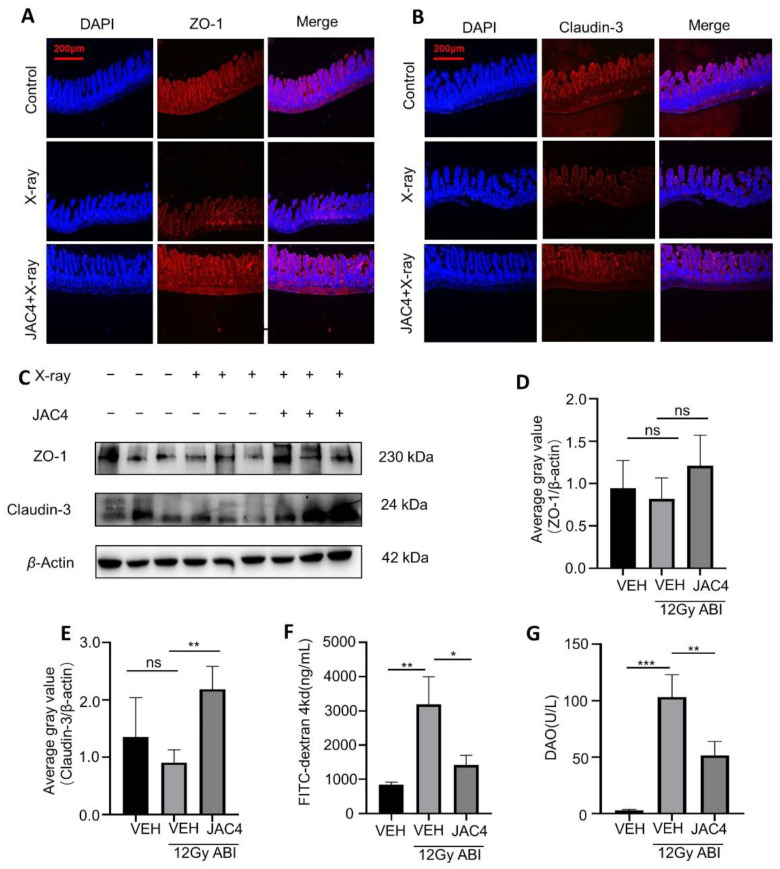
JAC4 treatment improved the intestinal mucosal barrier from radiation damage. (**A**,**B**) Represented images of immunofluorescence staining with Claudin-3 and ZO-1 antibodies in the jejunum on day 4 after ABI (n = 6). (**C**–**E**) Protein expression of Claudin-3 and ZO-1 in the jejunum on day 4 after ABI (n = 3). (**F**) The concentration of FITC-dextran in the plasma (n = 6). FITC-dextran was measured on day 4 after ABI. (**G**) DAO activity in the plasma. The results are expressed as mean ± SEM (* *p* < 0.05; ** *p* < 0.01; *** *p* < 0.001; ns: *p* > 0.05).

**Figure 3 antioxidants-11-01067-f003:**
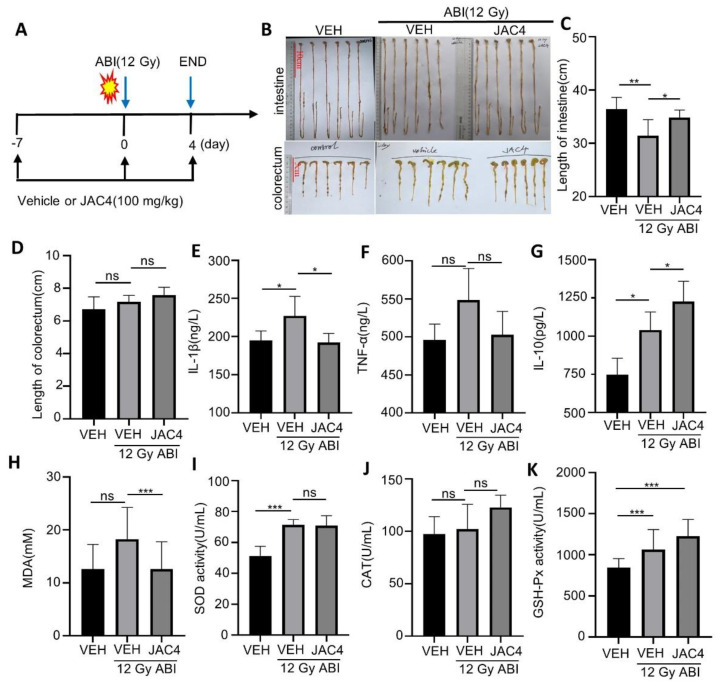
JAC4 inhibits X-ray-triggered inflammation and oxidative stress. (**A**) The schematic diagram of experimental design. C57BL/6 mice were pretreated with vehicle (VEH) or JAC4 for 7 days. Mice were then exposed to radiation (12 Gy). JAC4 treatment was continued for 4 days. Then, serum and small intestine tissues were collected. (**B**) Represented images of small intestine and colorectum; (**C**,**D**) quantitative statistics of small intestine and colorectum length; serum contents of IL-1β (**E**), TNF-α (**F**), and IL-10 (**G**) were determined on the fourth day after ABI (n = 6); serum contents of MDA (**H**), SOD (**I**), CAT (**J**), and GSH-Px (**K**) were determined on day 4 after ABI (n = 6); the results are expressed as mean ± SEM (n = 6) (* *p* < 0.05; ** *p* < 0.01; *** *p* < 0.001; ns: *p* > 0.05).

**Figure 4 antioxidants-11-01067-f004:**
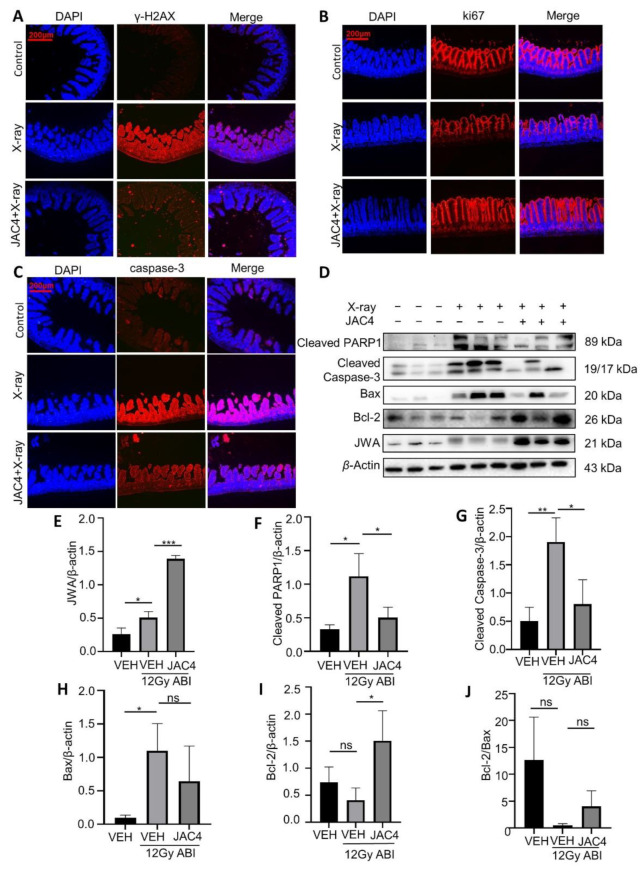
JAC4 attenuates DNA damage and apoptosis of the small intestine after ABI. (**A**–**C**) Represented images of immunofluorescence staining with γ-H2AX, Ki67, and caspase-3 antibodies in the jejunum on day 4 after ABI. (**D**–**I**) Protein expression of JWA, cleaved PARP1, Bax, and Bcl-2 in the jejunum after ABI. (**J**) Ratio of Bcl-2 to Bax gray value. Results are presented as mean ± SEM (n = 3) (* *p* < 0.05; ** *p* < 0.0; *** *p* < 0.001; ns: *p* > 0.05).

**Figure 5 antioxidants-11-01067-f005:**
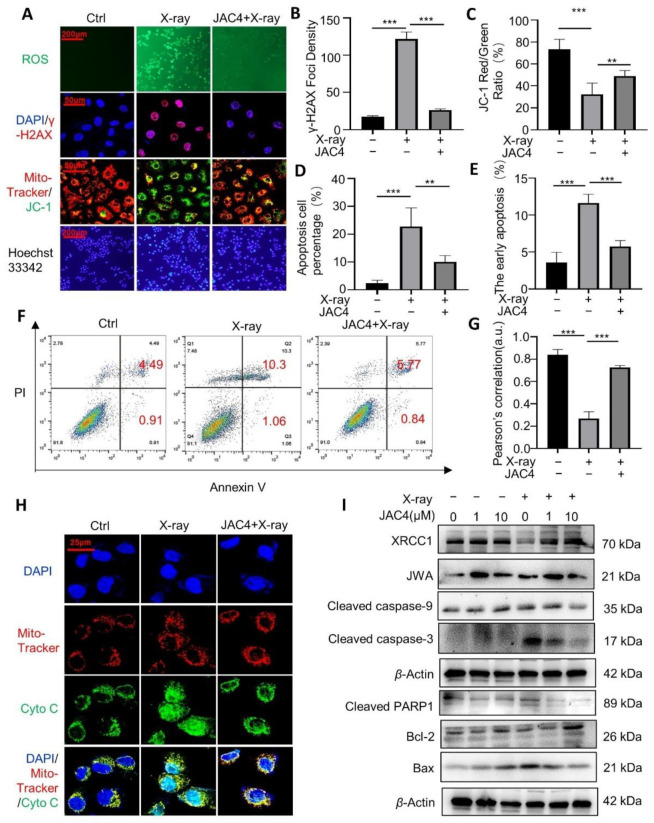
JAC4 attenuates X-ray-triggered oxidative stress and mitochondria damages in ICE-6 cells. (**A**) IEC-6 cells were pretreated with JAC4 (1 or 10 μM) or DMSO for 24 h and then exposed to X-ray (12 Gy). ROS was detected by DCFH probe, apoptosis was detected by Hoechst staining, the mitochondrial membrane potential was detected by JC-1 staining, and DNA damage was detected by γ-H2AX staining after 24 h. (**B**) Quantitative analysis of γ-H2AX foci density. (**C**) JC-1 red/green ratio of IEC-6 cells after treatment. (**D**) Quantitative analysis of apoptosis cell percentage by Hoechst staining. (**E**,**F**) Cells were stained with Annexin V-PI and analyzed by flow cytometry at 24 h after a dose of 12 Gy of radiation, and a histogram of apoptotic cells is expressed as a percentage of total cells. (**G**) Colocalization coefficient of cytochrome c and mitochondria of IEC-6 cells after treatment. (**H**) Corresponding fluorescence images of the release of cytochrome c from mitochondria in IEC-6 cells after treatment. (**I**) Western blot analysis was performed for XRCCI, JWA, cleaved caspase-3, cleaved caspase-9, cleaved PARP1, Bax, and Bcl-2 at 24 h after administration of 12 Gy of radiation. The results are expressed as mean ± SEM (n = 3) (** *p* < 0.01; *** *p* < 0.001; ns: *p* > 0.05).

**Figure 6 antioxidants-11-01067-f006:**
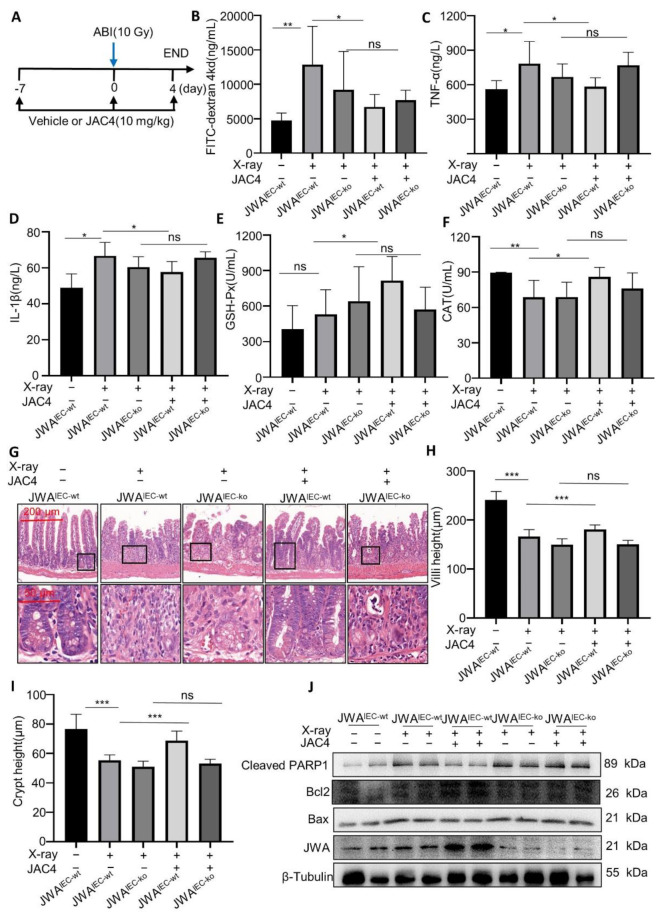
JWA is necessary for JAC4 protecting mice from X-ray exposure-induced intestinal barrier damage. (**A**) Both the JWA^IEC-ko^ and JWA^IEC-wt^ mice were administered JAC4 (10 mg/kg) or vehicle by gavage for 7 days, and the mice were then exposed to X-ray radiation (10 Gy); JAC4 treatment was continued for four days. (**B**) The concentration of FITC-dextran in the plasma. (**C**,**D**) The concentration of IL-1β and TNF-α in the plasma. (**E**,**F**) The activity of catalase (CAT) and glutathione peroxidase (GSH-Px) in the plasma. (**G**–**I**) H&E staining of small intestine and quantification of the villus height and crypt height. (**J**) Western blot analysis was performed for JWA, cleaved PARP1, Bax, and Bcl-2 after 10 Gy radiation. The results are expressed as mean ± SEM (n = 5 or 6) (* *p* < 0.05; ** *p* < 0.01; *** *p* < 0.001; ns: *p* > 0.05).

## Data Availability

The data supporting the conclusions of this article are presented within the article and its additional files.

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
