# Peer review of "JAC4 Protects from X-ray Radiation-Induced Intestinal Injury by JWA-Mediated Anti-Oxidation/Inflammation Signaling"

_antioxidants, 2022, doi:10.3390/antiox11061067_

Round 1

Reviewer 1 Report

The authors in their study pointed out that the problem of tissue damage due to radiation could be reduced.
The studio is very interesting, well organized.
I would like to have some clarifications from the authors.
Do the authors think the microbiota is affected or does it influence the damage?
Do the authors have information on the safety of JAC4?
Do the authors have information on toxicity?
In the Discussion, the Authors should highlight the possible clinical significance of their findings

Reviewer 2 Report

The present study was investigating the effects of JAW agonist compound 4 (JAC4) on anti-oxidation and anti-inflammatory against radiation-induced intestinal injury.  Theer are many typos, and some definition in the experiments are not very clear for readers to understand. 

  1. The definition of cre-loxp mice in your study is not followed the rule and are also different from the reference 13.
  2. In line 96-101, some you wrote JWAIEC and some you wrote JWAICE. Which one is correct? Besides, what does IEC ko mean?
  3. Please replace the H&E figures to more obvious (or bigger) photos.
  4. figure 3 should put the scale.
  5. As shown in figure 4D, JWA were not elevated by X-ray, and so do JWA wt mice under X-ray treatment (JWA protein expression were less than/ or no change when compared with JWA wt group) .
  6. In figure 5A, γH2AX should not present blue color. 
  7. In figure 5H, why cytochrome c locates in the nucei?
  8. Why the dosage of JAC4 is different in radition group (100 mg/kg) and JWA knock out group (10 mg/kg).
